# HP3D-V2V: High-Precision 3D Object Detection Vehicle-to-Vehicle Cooperative Perception Algorithm

**DOI:** 10.3390/s24072170

**Published:** 2024-03-28

**Authors:** Hongmei Chen, Haifeng Wang, Zilong Liu, Dongbing Gu, Wen Ye

**Affiliations:** 1Faculty of Electrical Engineering, Henan University of Technology, Zhengzhou 450001, China; chenhongmei@haut.edu.cn (H.C.); wanghaifeng@stu.haut.edu.cn (H.W.); 2School of Computer Science and Electronic Engineering, University of Essex, Colchester CO4 3SQ, UK; zilong.liu@essex.ac.uk (Z.L.); dgu@essex.ac.uk (D.G.); 3Division of Mechanics and Acoustics, National Institute of Metrology, Beijing 102200, China

**Keywords:** cooperative perception, 3D object detection, feature extraction, crossvehicle feature fusion

## Abstract

Cooperative perception in the field of connected autonomous vehicles (CAVs) aims to overcome the inherent limitations of single-vehicle perception systems, including long-range occlusion, low resolution, and susceptibility to weather interference. In this regard, we propose a high-precision 3D object detection V2V cooperative perception algorithm. The algorithm utilizes a voxel grid-based statistical filter to effectively denoise point cloud data to obtain clean and reliable data. In addition, we design a feature extraction network based on the fusion of voxels and PointPillars and encode it to generate BEV features, which solves the spatial feature interaction problem lacking in the PointPillars approach and enhances the semantic information of the extracted features. A maximum pooling technique is used to reduce the dimensionality and generate pseudoimages, thereby skipping complex 3D convolutional computation. To facilitate effective feature fusion, we design a feature level-based crossvehicle feature fusion module. Experimental validation is conducted using the OPV2V dataset to assess vehicle coperception performance and compare it with existing mainstream coperception algorithms. Ablation experiments are also carried out to confirm the contributions of this approach. Experimental results show that our architecture achieves lightweighting with a higher average precision (AP) than other existing models.

## 1. Introduction

The quest for accurate collaborative sensing solutions arises from the critical need to overcome the limitations encountered by single vehicle sensing systems, including challenges such as long-range occlusion and sparse sensor observations. In recent years, remarkable strides have been made in the areas of robotic sensing technologies and machine learning methods [1,2,3]. These advancements have notably bolstered the reliability of perception systems, with instances such as LiDAR point clouds [4,5,6,7] and the integration of multisensor data [8,9,10], thus showcasing exceptional performance within the domain of vehicle perception.

In the ever-evolving landscape of sensing technologies, high-precision sensing algorithms, despite recent advances, continue to grapple with formidable challenges [11]. LiDAR technology, known for its attributes such as light independence, precise spatial information, and resilience to occlusion [12,13], has become integral for autonomous navigation vehicles, thereby relying on its point cloud scanning to perceive their surroundings.

However, the accuracy of light detection and radar ranging, which are commonly used for acquiring the point cloud data of a scene containing the target, is often influenced by various factors. These factors include the platform and sensor accuracy, environmental interference, the reflective properties of the target, and the complexity of the target scene. Consequently, the accuracy of downstream vehicle perception algorithm models is reduced. Therefore, filtering and denoising the point cloud data (PCD) before feature extraction becomes an essential and crucial step in vehicle perception [14,15,16].

Furthermore, the development of multivehicle cooperative sensing algorithms poses significant challenges, particularly in the extraction of semantic features from sparse and voluminous unstructured data. Additionally, effectively processing, sharing, and fusing the feature information obtained from multivehicle sensing is a key issue in cooperative driving. Addressing these challenges is of paramount importance in the realization of a dependable vehicle cooperative system. Hence, in the realm of point cloud denoising, numerous filters have been devised by researchers to eliminate noisy data. These include the statistical filter [17], voxel filter [18], and radius filter [19]. However, these filters suffer from severe information loss, high computational complexity, and parameter dependence. In addition, existing 3D target detection algorithms mainly use mesh-based point cloud feature extraction methods, which can be broadly categorized into 3D voxel-based and 2D column-based methods. These methodologies adopt the traditional “encoder–neck” detection architecture [20,21,22,23,24,25,26,27,28]. Voxel-based methods [20,21,25,27,28] commonly involve segmenting the input point cloud into a regular 3D voxel mesh and establishing a geometric representation across various levels through an encoder utilizing sparse 3D convolutions. Following the encoder, the integration of multiscale features occurs through the neck module of a conventional 2D convolutional neural network (CNN) prior to the input entering the detection head. Conversely, voxel-based methods [22,23,24,26] involve the transformation of a 3D point cloud into a 2D pseudoimage within the BEV (bird’s-eye view) plane. Subsequently, these methods construct the neck network directly on a 2D CNN-based feature pyramid network (FPN) to facilitate the fusion of multiscale features. While voxel-based methods demonstrate strong detection performance, the challenge lies in effectively aggregating multiscale features with varying resolutions within the BEV space, which is primarily due to the constraints posed by 3D sparse convolutions within the encoder. On the contrary, the utilization of lightweight encoders for column feature learning in voxel-based methods often leads to reduced accuracy compared to voxel-based approaches. Moreover, the detection performance is further constrained by the combination of small-sized pseudoimages and the substantial size of the initial columns. These limitations hinder the overall effectiveness of the detection process. Against the aforementioned analysis, this paper takes point cloud data as the starting point and designs a point cloud filtering method, thereby aiming to improve the accuracy and reliability of the point cloud data. Meanwhile, we analyze the voxel and PointPillars methods in depth to solve the problem of lack of spatial feature interaction in the PointPillars-based feature extraction method. In addition, we have built a crossvehicle feature fusion module to capture the spatial relationships between features, which enables high-accuracy cooperative perception for 3D objective detection. Our main contributions are summarized as follows:A voxel grid-based statistical filter (voxel grid filter) is introduced in the preprocessing stage to improve the cleanness and reliability of the PCD.We present a feature extraction network structure for voxel point column fusion to solve the problem of the lack of spatial feature interaction in the point column-based feature extraction method, and we use maximum pooling to replace the feature splicing operation in the voxel-based method to realize the dimensionality reduction of the features and to generate a pseudoimage for the subsequent processing of pseudoimage features using a 2D CNN.We establish a cooperative perceptual feature fusion module to construct a feature compression and feature sharing network, and we introduce residual blocks to reduce the loss of information during network transmission. In addition, based on max and mean dimensionality reduction operators, we propose an adaptive feature fusion module to better capture spatial relationships between features, thus improving the accuracy of the model.

Our HP3D-V2V algorithm model was trained and validated on the OPV2V dataset in the default CARLA Towns and Culver City scenarios. First, we validated the superiority of laser radar point cloud collaborative perception over single-vehicle perception. Secondly, our algorithm achieved a higher AP compared to mainstream collaborative perception models. Finally, we conducted ablation experiments on the improved model proposed in this paper to validate its effectiveness.

The structure of this paper is organized as follows. In Section 2, we present a comprehensive analysis of the fusion strategy for cooperative perception. Section 3 describes our approach, including point cloud denoising, feature extraction, and fusion techniques. In Section 4, we provide insights into the dataset used, describe the experimental implementation details, and compare the results of our proposed method with the baseline approach. Finally, this paper is concluded in Section 5.

## 2. Related Works

The utilization of a point cloud vehicle-to-vehicle collaborative sensing pipeline involves encoding the raw point cloud data and subsequently decoding the features generated by the encoder to obtain the final sensing result. Existing collaborative approaches can be broadly categorized into three main types.

### 2.1. Early Collaboration

Early collaboration occurs in the input space where raw perceptual data is shared among vehicles. This approach involves aggregating the perceptual measurements from all vehicles to contribute to a comprehensive perspective. Consequently, each vehicle can process and perceive its surroundings based on a holistic view, as depicted in Figure 1a. In [29], Arnold proposed a cooperative 3D object detection approach using single-mode sensors, which integrates information from spatially diverse sensors distributed throughout the environment to alleviate the limitations of individual sensors. Meanwhile, Ref. [30] estimated the uncertainty of cooperative object detection for CAVs and introduced a novel method called Double-M quantization, which is capable of capturing epistemic uncertainties. Although early collaborative models have been shown to effectively address occlusion and limitations in single-vehicle perception, the sharing of raw sensor data requires extensive communication and is susceptible to network congestion due to large data payloads, thus limiting its practical applicability in many scenarios.

### 2.2. Intermediate Collaboration

Intermediate collaboration takes place in the intermediate feature space, where each individual agent transmits intermediate features generated based on predictive models. These features are then fused, and each agent decodes the fused features to generate perceptual results, as illustrated in Figure 1b. In essence, the representative information can be compressed into these features. Intermediate collaboration offers a more efficient communication bandwidth compared to early collaboration, and it has been shown to enhance perception compared to late collaboration. F-Cooper [32] proposed a feature-level fusion scheme that utilizes the maximum value in overlapping regions to represent intermediate features. The makers of Opv2v [33] constructed a comprehensive benchmarking framework and introduced a novel focused intermediate fusion pipeline for aggregating information from multiple connected vehicles. The makers of V2VNet [34] employed graph neural networks to aggregate shared neural features for joint detection and prediction. The makers of V2X-ViT [35] explored the use of window attention and heterogeneous self-attention to achieve vehicle-to-everything cooperation in visual transformers, thus designing a heterogeneous multiagent attention module (HMSA) and a multiscale window attention module (MSwin) for heterogeneous V2X perception. The makers of V2VFormer [36] adopted a transformer-based collaborative approach that dynamically performs semantic interaction based on positional correlation, thus serving as a lightweight plug-and-play module. The makers of CORE [37] enabled efficient reconstruction of observations through a compressor, a lightweight attentional collaboration component, and a reconstruction module, thereby providing clear and effective oversight for improving the efficiency of perception tasks. However, the intermediate cooperative perception approach faces two major challenges. Firstly, it involves selecting the most beneficial and compact features from the original measurements for transmission. Secondly, it aims to maximize the fusion of features from other vehicles to enhance the perceptual capabilities of each vehicle.

### 2.3. Late Collaboration

The postcollaborative approach, which involves sending detection outputs and fusing received suggestions into consistent predictions, operates in the output space, such as bounding boxes in 3D target detection. This enables the fusion of perceptual results generated by individual agents, as depicted in Figure 1c. The makers of UMC [38] utilized multiresolution technology to enhance the communication, collaboration, and reconstruction processes, thereby incorporating a novel trainable multiresolution and selective region mechanism in communication and integrating multiresolution collaborative features in reconstruction. In addition, Ref. [39] investigated the temporal and spatial alignment of shared detection objects, thereby proposing to utilize nonpredictive sender states for transformations in order to ignore the motion compensation of the sender. However, the late collaboration approach has certain limitations. Firstly, it is highly sensitive to the localization errors of the agent, which can arise from incomplete local observations and result in significant estimation errors and noise. Secondly, the late collaboration approach heavily relies on the sensor data of a single vehicle and functions optimally only when all agents share their sensing results, thus limiting its direct applicability.

## 3. HP3D-V2V Algorithm

This paper proposes a high-precision 3D target detection algorithm for vehicle-to-vehicle cooperative perception, thus building upon the OPV2V framework, and the overall structure is shown in Figure 2. The algorithm processes point cloud data through six steps:Filtering the input point cloud data to enhance its quality.Utilizing voxel column fusion to perform feature coding on the filtered point cloud, thus resulting in a pseudoimage representation known as the pillar feature network (PFN).Extracting multiscale features from the PFN using a feature pyramid network (FPN), thereby enabling the extraction of intermediate features.Performing intervehicle data sharing, where the intermediate feature map of the cooperative autonomous cehicle (CAV) is projected onto the self-vehicle coordinates.Conducting intervehicle feature fusion to generate a combined feature map that integrates information from multiple vehicles.Performing 3D object detection to output a bird’s-eye view (BEV) representation of the detected 3D targets.

### 3.1. Point Cloud Denoising

Given the complexity of the environment in vehicle perception tasks, LiDAR point cloud data (PCD) usually has a large amount of nonvehicle noise, such as ground noise, wall point noise, and sensor noise [40]. To effectively eliminate these noises, this paper proposes a voxel grid-based statistical filter scheme. The specific steps are as follows: First, the original point cloud data are voxelized, and a KD tree is constructed for each voxel to improve the efficiency of the nearest neighbor search. Grid-based principal component analysis (GPCA) is then employed to compute normal vectors, which serve as salient features. An unsupervised method is utilized for the rough segmentation of noise based on these computed normal vectors. To further enhance the denoising effect, this paper introduces a k-nearest neighbor (KNN)-based correction scheme. This scheme determines whether each point should be retained by calculating the average distance to its k-nearest neighbors and comparing it with a preset threshold. Figure 3 illustrates this process. By employing these techniques, the proposed method effectively addresses the issue of nonvehicle noises in LiDAR point cloud data, thereby leading to improved denoising results. Firstly, the three-bit point cloud information is divided into an equally spaced voxel grid, and a KD tree is built for each voxel to facilitate the KNN search. Assume that the input point cloud data are 
$S\in R^{n\times 3}$
, which contain three-dimensional space with ranges *D*, *H*, and *W*, along the *Z*, *Y*, and *X* axes, respectively. Accordingly, each voxel size is defined as 
$v_{D}$
, 
$v_{H}$
, and 
$v_{W}$
, and the dimensions of the 3D voxel grid are obtained as 
$D^{\prime }=D/v_{D}$
, 
$H^{\prime }=H/v_{H}$
, and 
$W^{\prime }=W/v_{W}$
, respectively.

Secondly, the PCD are downscaled using GPCA. The covariance matrix of the input point cloud data *S* is decomposed using singular value decomposition (SVD) to obtain the corresponding feature vectors. The first two feature vectors are chosen to form the dimension reduction matrix for input *S*.

(1)
$$S_{q}=S\Delta_{T}.$$


The dimension reduction matrix 
$\Delta_{T}\in R^{3\times 2}$
 transforms the 3D point set S into the 2D point set 
$Sq=x_{i},y_{i},i=1,...,n$
. After dimension reduction, the obtained 2D data are meshed with *l* meshes. The resolutions along the *x* and *y* directions are denoted by 
$b_{x}$
 and 
$b_{y}$
, respectively.

(2)
$$\left\{\begin{array}{c} b_{x}=\frac{x_{\max}-x_{\min}}{l}\\ b_{y}=\frac{y_{\max}-y_{\min}}{l} \end{array}\right.$$


(3)
$$\left\{\begin{array}{c} h_{xi}=round\left(\left(x_{i}-x_{\min}\right)/b_{x}\right)\\ h_{yi}=round\left(\left(y_{i}-y_{\min}\right)/b_{y}\right) \end{array}\right.$$

where each point has a code 
$\left(h_{xi},h_{yi}\right)$
 restricted to the range 1 to *l* as follows:
(4)
$$h=\left\{\begin{array}{c} 1,\;h=0\\ l,\;h>l \end{array}\right.$$


The number of points 
(i,j)
 projected onto the grid using GPCA is denoted by 
$k_{ij}$
. To partition the point cloud into two parts, a threshold parameter 
$k_{\delta }$
 is defined as in the following equation:
(5)
$$x_{ij}\in \left\{\begin{array}{c} S_{f},k_{ij}<k_{\delta }\\ S_{w},k_{ij}\geq k_{\delta } \end{array}\right.$$

where 
$x_{ij}$
 denotes the points 
(i,j)
 in the grid, 
$S_{f}$
 is the set of vehicle information points containing ground noise and sensor noise, and 
$S_{w}$
 is the set of wall point noise points, mainly defined for vertical structures such as trees, walls, and obstacles. 
$S_{f}$
 is used as an input to the subsequent algorithms, which utilize an unsupervised approach to coarsely segment the point cloud based on the computed normal vectors.

To efficiently extract feature information from the point cloud, we first construct the KD tree of 
$S_{f}$
 to locate the *K* nearest neighbors 
$P_{i}$
 for each point in the tree. Next, we compute the minimum sum of distances between a plane and its nearest neighbors and extract the normal vector of the plane as the feature for the corresponding point. By utilizing PCA, the normal vector of the PCD can be swiftly derived as follows:

For each of the *K* sets of nearest neighbors, the mean and deviation errors are computed.

(6)
$$\mu_{j}=\frac{1}{k}\sum_{j=1}^{k}x_{j}$$


(7)
$${\tilde{x}}_{j}=x_{j}-\mu_{j}$$

where 
$\mu_{j}$
 represents the mean of the nearest neighbor, while 
$x_{j}$
 denotes the difference in distance between the point and the mean 
$\mu_{j}$
. The corresponding deviation matrix *C* is defined as follows:
(8)
$$C=[{\tilde{x}}_{1},{\tilde{x}}_{2},...,{\tilde{x}}_{k}]$$


An SVD decomposition of the 
$CC^{T}$
 will be obtained as follows:
(9)
$$U\Sigma V^{T}=CC^{T}$$


The normal vector of the corresponding point 
$v_{i}$
 is determined by the eigenvector associated with the smallest eigenvalue in *U*. Employing an unsupervised approach, the point cloud is roughly segmented into vehicular and nonvehicular regions based on the computed normal vector.

Given that the point set 
$S_{f}$
 may contain ground noise and sensor noise, the threshold angle between normal vectors serves as a critical metric for segmentation denoising. The primary steps for denoising 
$S_{f}$
 are outlined as follows. Begin by selecting a random point 
$v_{i}$
 from the initial dataset 
Ω
. Then, compare the angle between the normal vectors of 
$v_{i}$
 and all other points 
$v_{j}$
 as follows:
(10)
$$\theta =\frac{v_{i}\cdot v_{j}}{∥v_{i}∥\times ∥v_{j}∥}$$


Specify a threshold 
δ
. If the angle of a 
$v_{j}$
 is less than 
δ
, classify 
$v_{j}$
 as belonging to the same category as 
$v_{i}$
. Utilize the remaining points as the updated initial dataset 
Ω
, thereby iterating this process until all noise points are successfully eliminated.

The described process is highly effective, thus utilizing the inclusion angle of the normal vector as a key indicator. However, in complex traffic scenarios, this approach may introduce errors. To minimize such errors, a further correction method based on k-nearest neighbors (KNN) is employed. For each point 
$P_{i}$
 in the point cloud data 
$S_{f}$
, the proximity of each point in 
$S_{f}$
 is evaluated by calculating its average distance to its nearest neighbor using the KNN method, which can be expressed as follows:
(11)
$$AvgDist\left(i\right)=\frac{1}{K}\sum_{j=1}^{K}Dist(P_{i},P_{j})$$

where 
AvgDist(i)
 denotes the average distance between the *i*th point and its *K* nearest neighbors, 
$P_{i}$
 denotes the coordinates of the *i*th point, and 
$Dist(P_{i},P_{j})$
 denotes the Euclidean distance between points 
$P_{i}$
 and its nearest neighbor 
$P_{j}$
.

The calculated average distance, denoted as 
AvgDist
, is then compared to a predefined threshold. If 
AvgDist
 is found to be below the threshold, the corresponding point is identified as a noise point and should be filtered out. Conversely, if 
AvgDist
 exceeds the threshold, the point is considered to be valid and is retained in the denoised point cloud. By implementing these steps, the proposed method effectively removes noise from the LiDAR point cloud, thereby leading to improved accuracy and reliability in vehicle perception.

### 3.2. Feature Learning Network

To address challenges related to occlusion and scale variation, bird’s-eye view (BEV) methods have gained popularity for 3D object detection. Two commonly employed techniques for projecting the point cloud onto the BEV plane are voxelization and pillarization. The voxelization method involves extracting features through 3D convolution across the height 
(H)
, width 
(W)
, and depth 
(D)
 dimensions, with *D* representing the height dimension and *C* denoting the feature channel. Following downsampling, the resulting features are reshaped into BEV features of size 
$\left(H^{\prime },W^{\prime },D^{\prime }\times C^{\prime }\right)$
. This process allows for capturing volumetric information and maintaining feature resolution across all dimensions.

While voxelization methods excel at preserving fine-grained features, they often rely on computationally intensive 3D convolutions. On the other hand, pillar methods [41] offer higher computational efficiency by simplifying the point cloud feature extraction process. However, they suffer from information loss in the height dimension. When neighboring points are assigned to different columns in 3D space, these points only contribute to the feature extraction within their respective columns. As a result, the feature correlation between these points is overlooked, which hampers the extraction of local features from the point cloud. To address these issues, we construct voxel pillars on voxel feature maps and encode them to generate BEV features, thereby addressing the issue of spatial feature interaction lacking in PointPillars [22] methods and enhancing the semantic information of extracted features. Additionally, we employ a max pooling instead of the feature concatenation operations used in VoxelNet [28], thus resulting in more compact BEV feature maps and avoiding two-dimensional convolution operations on invalid feature channels. The specific improvements are illustrated in Figure 4, which provides a visual representation of the changes made to enhance the feature extraction process.

First, the point cloud is voxelized, and the 3D index of each point is calculated to convert the point level features to voxel level features with dimensions 
(H,W,D,C)
. The voxel level features are extracted using a voxel feature encoding (VFE) module. This module consists of fully connected layers, maximum pooling, and point-by-point splicing operations; the details are shown in Figure 5. The input point cloud is converted into voxel-level features by applying multiple VFE modules and performing element-level maximum pooling operations. In addition, voxels of different sizes can be used as input and fed into the VFE modules to obtain pseudoimage features of different sizes.

The features are extracted by the VFE module to obtain a tensor of dimension 
$\left(H^{\prime },W^{\prime },D^{\prime },C^{\prime }\right)$
. Then, column construction is performed on the output voxel feature map, and the voxel features within the column are encoded and pooled by combining two dimensions 
(D×C)
 for the scattering operation to obtain a tensor with the feature volume of 
$\left(H^{\prime },W^{\prime },D^{\prime }\times C^{\prime }\right)$
, where 
$D^{\prime }\times C^{\prime }$
 denotes the number of feature channels after dimensionality reduction. Finally, the 3D features after the pillar are reorganized through the Scatter module used in the literature [22] and assigned to the corresponding pseudoimage pixel positions according to their spatial locations to form a pseudoimage in the form of 
$\left(H^{\prime },W^{\prime },D^{\prime }\times C^{\prime \prime }\right)$
, which is done in order to avoid the subsequent computation of complex 3D convolution.

By integrating the advantages of voxelization and pillarization, our novel approach seeks to overcome the drawbacks associated with each method individually. This hybrid strategy enables a more effective transformation of point clouds into BEV features, thus facilitating improved object detection and localization in 3D perception tasks.

### 3.3. Backbone

In this paper, a feature pyramid network based on standard convolutional and transposed convolutional layers is proposed for the “near-dense and far-sparse” characteristics of vehicle LiDAR point cloud data, as shown in Figure 6. The backbone of the network consists of three key components: a top-down network, a transposed convolutional network that performs upsampling, and a feature aggregation network with different layers. The top-down network learns features at a smaller resolution, thus passing information through a top-down path for more global features. The transposed convolutional network performs an upsampling operation to restore the feature map size to the original size. A different layers feature aggregation network is used to fuse features from different layers to obtain a more global and semantic feature representation.

We opt for the direct concatenation method to aggregate feature maps from different layers based on several considerations. Firstly, this approach maintains information integrity between feature maps of varying layers, thereby mitigating information loss and enhancing feature expression capability. Secondly, the direct concatenation method boasts low computational complexity, thereby requiring no additional computational operations and enhancing both the training and inference speeds of the network. Lastly, this method simplifies the network structure, thereby reducing complexity and the risk of overfitting.

Convolutional operations in a network can be described by a series of blocks, Block 
(S,L,F)
, each consisting of multiple 
3×3
 2D convolutional layers with the same number of output channels. Specifically, each block consists of *L*

3×3
 convolutional layers with *F* output channels, and each layer is appended with BatchNorm normalization and a ReLU activation function after the convolution operation. The size of the input pseudoimage can be varied by adjusting parameters such as the step size *S*, padding, and convolution kernel size. The final output features are the concatenation of all the features from different step sizes. Compared with upsampling methods such as bilinear interpolation and bicubic interpolation, the convolution kernel parameters of transposed convolution can be updated and adjusted using backpropagation during the training phase of the model, thus making the sampling parameters more reasonable.

### 3.4. Multivehicle Information Fusion Pipeline

In this paper, we propose an intermediate fusion pipeline for the problems of prefusion bandwidth consumption, postfusion localization error sensitivity, and information interaction delay in vehicle-to-vehicle cooperative sensing, as shown in Figure 2. The method aims to effectively control bandwidth consumption and capture the interactions between the features of the neighboring connected vehicles in order to improve the sensing accuracy, and the core modules are as follows.

#### 3.4.1. Data Sharing and Feature Extraction

In this module, each cooperative autonomous vehicle (CAV) broadcasts its own relative attitude and external information to construct a spatial directed graph, where each node represents a cooperative autonomous vehicle (CAV) within the communication range, and the edges denote the communication channels between the nodes. Subsequently, each CAV projects its own point cloud data onto the autonomous (Ego) vehicle’s LiDAR coordinate system and performs feature extraction based on the projected point cloud data. By designing the feature extraction in Section 3.2, the CAV is able to extract distinguishable and information-rich features from the point cloud data.

#### 3.4.2. Feature Compression and Sharing

We introduce an encoder–decoder architecture tailored for the compression and decompression of shared feature information. During the compression stage, we employ the variational image compression algorithm, as proposed by Ball et al. [42], to efficiently compress features. Using a convolutional network, we compress the middle layer feature representation and subsequently apply quantization and lossless encoding techniques by utilizing entropy coding. In the decompression phase, the compressed information undergoes decoding via multiple inverse convolutional layers [43]. This process reconstructs the original feature representation, which is then transmitted to the feature aggregation module. Consequently, our approach minimizes communication overhead while ensuring efficient feature transfer. This facilitates the provision of accurate and informative features to each vehicle, thereby enhancing perception and decision-making capabilities.

#### 3.4.3. Crossvehicle Feature Fusion (CVFF)

The CVFF module integrates compressed features from various vehicles to derive global perceptual information. To fuse feature maps, commonly employed intuitive dimensionality reduction operators such as max [32] or mean are utilized. These operators, involving max pooling and average pooling operations on the channel axes, respectively, generate fused feature maps denoted as 
$F_{fusion}\in R^{1\times C\times H\times W}$
. In this paper, we amalgamated the two methods to flexibly utilize the spatial features, as depicted in Figure 7. This method executes adaptive feature fusion based on the spatial features derived from both maximum pooling and average pooling. Initially, the input feature map 
$F\in R^{n\times C\times H\times W}$
 is decomposed to produce 
$F_{max}\in R^{1\times C\times H\times W}$
 and 
$F_{avg}\in R^{1\times C\times H\times W}$
, thus representing the outcomes of maximum pooling and average pooling, respectively. These two feature maps are concatenated to form a 4D tensor 
$F_{spatial}\in R^{2\times C\times H\times W}$
, thereby encapsulating both types of spatial information from the original concatenated intermediate feature maps. Subsequently, a 3D convolution with a ReLU activation function is employed to selectively downscale the features, thereby yielding 
$F_{fusion}\in R^{2\times C\times H\times W}$
. This spatially adaptive feature fusion approach facilitates dynamic utilization of the spatial features based on the specific task, thus downsizing them while preserving key information. Consequently, this methodology better captures the spatial relationships between features.

### 3.5. Loss Functions

Similar to other point column-based approaches in the literature, the proposed 3D target detection network utilizes the same localization loss function as proposed in [28], thereby using the SmoothL1 function [44] to compute the position loss as follows:
(12)
$$L_{loc}=\sum_{i}^{N_{a}}L_{reg}\left(\delta_{i},t_{i}\right)$$


(13)
$$\begin{array}{cc} L_{reg}\left(\delta_{i},t_{i}\right)= & \sum_{j\in \{x,y,z,l,w,h\}}L_{sm}\left(\delta_{ij}-t_{ij}\right)\\ \; & +\sum_{j\in \{\theta \}}L_{sm}\left(\sin\left(\delta_{ij}-t_{ij}\right)\right) \end{array}$$


(14)
$$L_{sm}\left(x\right)=\left\{\begin{array}{c} 0.5x^{2},\;if\left|x\right|<1,\\ \left|x\right|-0.5,\;x<-1\cup x>1 \end{array}\right.$$

where 
$N_{a}$
 is a constant representing the total number of anchor frames, and 
$\delta_{i}$
 and 
$t_{i}$
 are the predicted and true values of the vehicle target, respectively, both of which include seven dimensions 
(x,y,z,w,l,h,θ)
. For the categorization branch of the detection output, focus loss is used to deal with the unbalanced target category loss of positive and negative samples, as in shown in the following equation:
(15)
$$L_{clc}=-\alpha_{a}{\left(1-p_{a}\right)}^{\gamma }\log\left(p_{a}\right)$$



$P_{a}$
 is the category probability: the closer 
$P_{a}$
 is to 1 means the higher the probability that the current target is a vehicle; the hyperparameter 
α
 is a balancing factor used to balance the proportion of positive and negative samples. This paper sets 
α
 to 0.25; 
γ
 is the difficult and easy samples adjustment factor, which is designed to make the model pay more attention to difficult-to-classify samples and wrongly classified samples, and this paper sets the 
γ
 value to 2.

In summary, the total loss is expressed as follows:
(16)
$$L_{total}=\frac{1}{N_{pos}}\left(\beta_{1}L_{loc}+\beta_{2}L_{clc}\right)$$

where 
$N_{pos}$
 denotes the number of positive anchor frames, and the loss weights 
$\beta_{1}$
 and 
$\beta_{2}$
 are 1.0 and 2.0, respectively.

## 4. Experiments

Our proposed algorithm was evaluated using the OpenV2V (ICRA2022) public dataset. The dataset setup and partitioning details are explained in Section 4.1, while the implementation specifics and evaluation metrics are outlined in Section 4.2. To assess the performance of our algorithm, in Section 4.3, our proposed HP3D-V2V algorithm compares the benchmark model with the mainstream algorithm. Additionally, we conducted ablation experiments in Section 4.4 to systematically evaluate the effectiveness of the proposed HP3D-V2V algorithm presented in this paper.

### 4.1. Dataset and Split

OPV2V stands as the premier large-scale open dataset designed for V2V (vehicle-to-vehicle) communication awareness [33]. This dataset comprises aggregated sensor data gathered from numerous interconnected self-driving vehicles, thereby encompassing 73 scenarios, six road types, and nine cities. The data collection was executed through the utilization of OpenCDA’s cooperative driving cosimulation framework [45] and the CARLA simulator [46]. Each scene within the dataset spans a duration of 16.4 s and involves a 64-channel LiDAR capture producing 1.3 million points per second. We used 2189/631/947 frames for training/validation/testing, respectively, to ensure the feasibility on limited equipment.

### 4.2. Implementation Details

#### 4.2.1. Device Information

In this experiment, the network model was built using the PyTorch framework and deployed on an Intel(R) i7-11800H CPU (Santa Clara, CA, USA) and RTX 3080 GPU (NVIDIA, Santa Clara, CA, USA) for parameter training and result validation. In addition, the Open3D tool was used to visualize the point cloud and draw 3D target frames for the vehicle objects.

#### 4.2.2. Metrics

We used the common settings in [25,28] to train the model by selecting LiDAR points as regions of interest along the *X*, *Y*, and *Z* axes, respectively, in the following ranges: (−140.8 m, 140.8 m), (−40 m, 40 m), and (−3 m, 1 m). We set the broadcasting range between the Cavs to 70 m. In training, we used matching thresholds of 0.6 and 0.45 for positive and negative samples, respectively. The matching IoUs between the bounding box and anchor points were calculated according to their nearest horizontal rectangles in the BEV. The length, width, and height of the anchor box used to detect the Cavs were 3.9 m, 1.6 m, and 1.56 m, respectively, the range of rotation angles of the anchor box was [0, 90], and the number of anchor boxes was two. The average precision (AP) at crossunion (IoU) thresholds of 0.5 and 0.7 was used to evaluate the different models.

#### 4.2.3. Model Details

We used a 3D voxel mesh to represent the 3D world in a binary representation, and we assigned a positive label to each voxel if it contained point cloud data. The voxel size was set to [0.4, 0.4, 0.4], and each voxel contained at most 32 points. The maximum number of voxels in the training set was 32,000. The VFE section was normalized and used absolute coordinates, and the number of output channels was 64. The number of output features in the PointPillars Scatter section was 64. The backbone section consisted of three layers, with their respective number of layers, steps, and channels being [3, 5, 8], [2, 2, 2], and [64, 128, 256]. The upsampling step was [1, 2, 4], and the number of channels in the upsampling module was [128, 128, 128]. Finally, for each pillar, the model predicted the classification labels using a classification header and predicted the classification labels using a regression header that predicted the seven degrees of freedom parameters of its nearest box.

#### 4.2.4. Training

We trained for 30 epochs using the Adam optimizer to update the model parameters. The batch size, learning rate, and weight decay were 2, 0.002, and 0.001, respectively, with a momentum range of [0.85, 0.95]. We used a multistep learning rate scheduler to dynamically adjust the learning rate, with a step size set to [20, 30] and a decay rate of 0.1. In the inference phase, we filtered out low-confidence bounding boxes by a threshold of 0.3. The IoU threshold for nonmaximum suppression (NMS) was 0.2.

#### 4.2.5. Data Augmentation

We applied three data augmentation methods: random flip, random rotation, and random scaling. The random flip method flips along the *x* axis, the random rotate method rotates in the world coordinate system in a given angular range of 
[−π/4,π/4]
, and the random scaling method scales in a scale range of [0.95, 1.05]. The enhanced visualization is shown in Figure 8. In addition, some objects were randomly selected from the training data and injected into the training samples.

### 4.3. Comparison Experiments

#### 4.3.1. Results

The performance of our proposed high-precision intermediate fusion collaborative sensing algorithm was assessed using the OPV2V dataset, and the corresponding results are presented in Table 1. To establish a comprehensive comparison, we evaluated our model against various fusion approaches, including the baseline model [33] that encompasses no fusion, early fusion, and late fusion. Additionally, we compared the proposed model with mainstream algorithms for collaborative perception based on intermediate collaboration [4,32,33,34].

We evaluated the performance of our model on the Default Towns and Culver City test sets of OPV2V, as shown in Figure 9. By examining Figure 9a,b, it is evident that both the AttFuse and V2VNet models exhibited confusion in handling bushes and structures, thus misclassifying them as vehicles. This confusion may stem from the visual similarity between these objects and vehicles, particularly in blurry or occluded conditions. Furthermore, through the areas labeled in the figure, we can also clearly observe that in terms of long-distance detection, the AttFuse model and the V2VNet model failed to adequately capture the detail information, and there were cases of missed vehicle detection.

By examining the prediction results in the labeled box section of the figure, our model was shown to demonstrate effective discrimination, thereby successfully avoiding the misidentification of shrubs and structures as carriers. Moreover, it exhibited enhanced accuracy when dealing with long-range vehicles returning to the bounding box. Regardless of occluded or distant vehicles, our model leverages the assistance of other CAVs to perceive occluded objects and achieve superior overall perception results.

#### 4.3.2. Discussion

From Table 1, it is evident that the collaborative perception model outperformed single-vehicle perception without fusion. Additionally, models based on intermediate fusion performed better than those utilizing early or late fusion. Our HP3D-V2V algorithm demonstrated high-precision detection on the OPV2V dataset, thereby achieving approximately or exceeding an 8.0% AP@0.7 on the CARLA Towns and Culver City datasets. Both the mainstream models and our proposed HP3D-V2V exhibited commendable detection performance on the dataset; however, our model showed a performance increase of 10.1% and 8.3% in AP@0.7 on the CARLA Towns and Culver City datasets, respectively. The experimental results validate that our point cloud denoising method enhances the model’s adaptability to the environment, thus reducing false positive detections in blurry or occluded scenes. Additionally, the feature extraction network and CVFF module demonstrated significant advantages, thus performing better in three-dimensional bounding box regression, particularly in long-distance detection.

### 4.4. Ablation Studies

In order to evaluate the validity of our proposed model, we selected the representative methods SECOND and PointPillars, which are voxel-based and point pillar-based in the baseline model, to perform ablation experiments with our proposed three improved points, and the evaluation results are shown in Table 2.

During the point cloud preprocessing stage, we introduced a voxel mesh-based statistical filter to obtain more reliable point features, which resulted in an 8.9% improvement in detection accuracy in 3D AP@0.7. As depicted in Figure 10b, this approach effectively mitigated the issue of misidentifying shrub structures as vehicles. To address the limitations of spatial feature interaction in PointPillars-based feature extraction methods while preserving finer-grained features, we utilized a voxel point pillar fusion (VPCF) scheme in the feature extraction phase. By examining Figure 10c, we can observe that the incorporation of the VPCF module significantly reduced the number of missed vehicles at long range. This improved the accuracy by 6.1% and 10.5% in AP@0.5 and AP@0.7, respectively. Note that the SENCOND method was not tested here, since the output of our feature extraction module is in the form of three-dimensional features. Finally, our proposed CVFF exceled at capturing representative features through feature interaction, thereby leading to a notable improvement in the AP, as demonstrated in Figure 10d. Moreover, our model exhibited enhanced accuracy in vehicle regression bounding box estimation.

## 5. Conclusions

In this paper, we investigated cooperative perception utilizing LiDAR point cloud data and proposed a method for high-precision 3D object detection in V2V scenarios, thereby aiming to overcome the challenges presented by complex road conditions that hinder detection accuracy. Initially, to ensure the reliability of the data for the feature extraction module, we devised a voxel grid-based statistical filter to denoise the point cloud. Subsequently, we designed a feature extraction module based on voxel and point column fusion to enhance the semantic information of the spatial feature interaction and feature extraction. Furthermore, we established an intermediate fusion approach for adaptively integrating spatial features across vehicles. Comparative evaluations against various mainstream cooperative perception algorithms demonstrate the superior detection accuracy achieved by our proposed algorithm. Furthermore, the efficacy of the proposed denoising method, VFE_VP, and CVFF modules has been further substantiated through ablation experiments.

To advance the efficiency and accuracy of autonomous driving and intelligent transportation systems, our future endeavors will explore multimodal fusion strategies and diverse point coding or detection networks to enhance overall system performance.

## Figures and Tables

**Figure 1 sensors-24-02170-f001:**
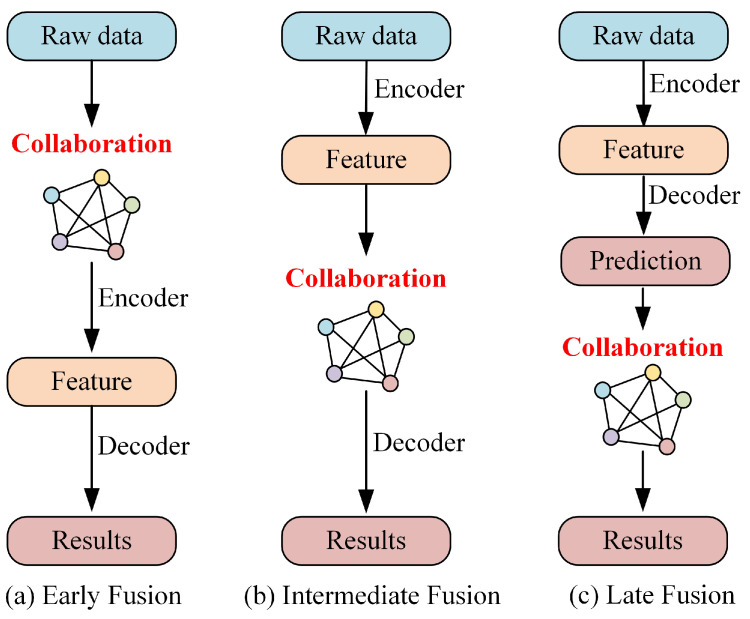
Diagram of the three types of collaboration strategies [31]. (**a**) Precoordinated vehicle feature fusion process. (**b**) Midphase feature fusion process of the cooperative vehicle. (**c**) Late feature fusion process of the cooperative vehicle.

**Figure 2 sensors-24-02170-f002:**
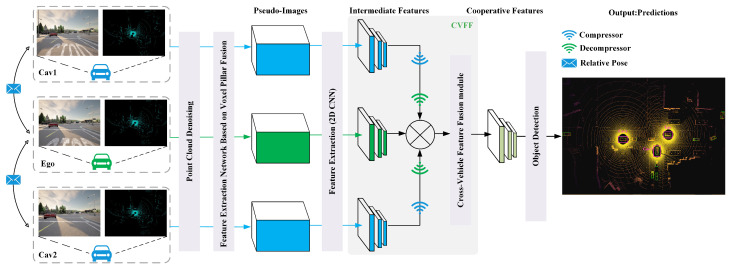
Proposed algorithm architecture for high-precision 3D target detection vehicle-to-vehicle cooperative perception.

**Figure 3 sensors-24-02170-f003:**
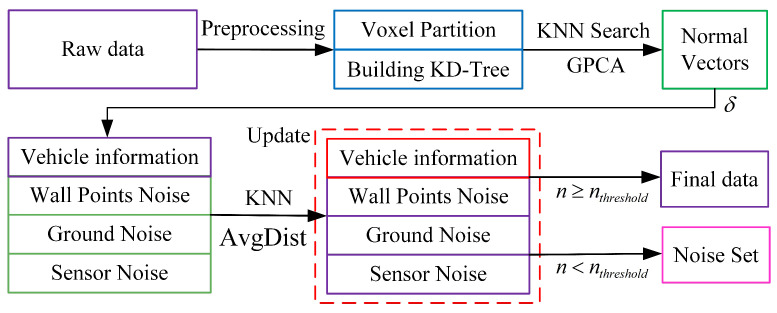
3D LiDAR point cloud denoising flow chart.

**Figure 4 sensors-24-02170-f004:**
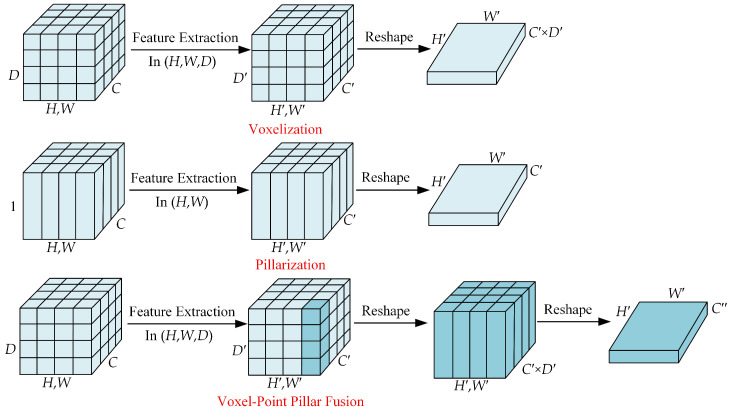
Comparison of feature extraction strategies.

**Figure 5 sensors-24-02170-f005:**
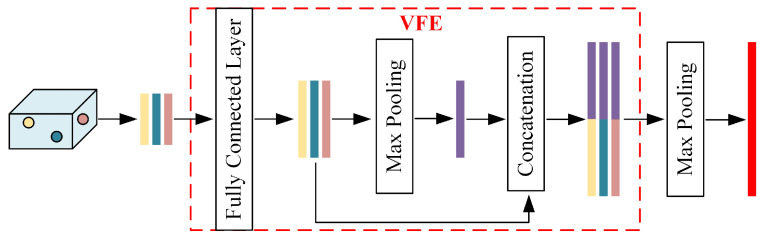
Structure of the voxel feature extraction network.

**Figure 6 sensors-24-02170-f006:**
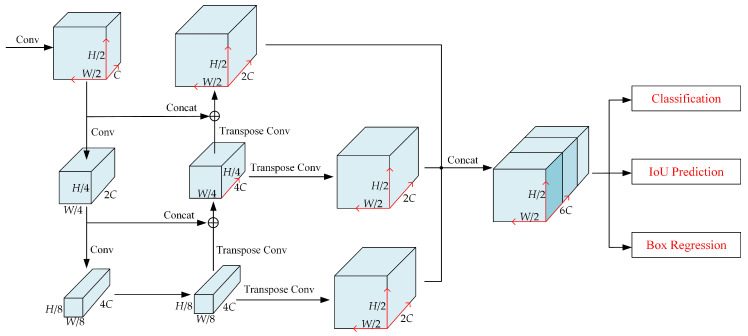
Feature pyramid backbone network with multibranch target detection head structure.

**Figure 7 sensors-24-02170-f007:**
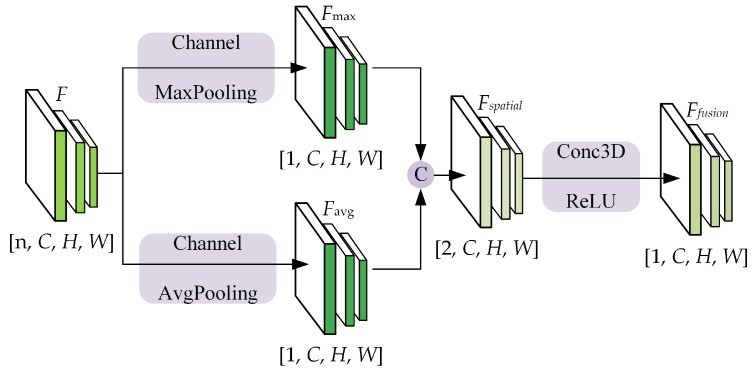
Crossvehicle feature fusion networks for intermediate feature.

**Figure 8 sensors-24-02170-f008:**
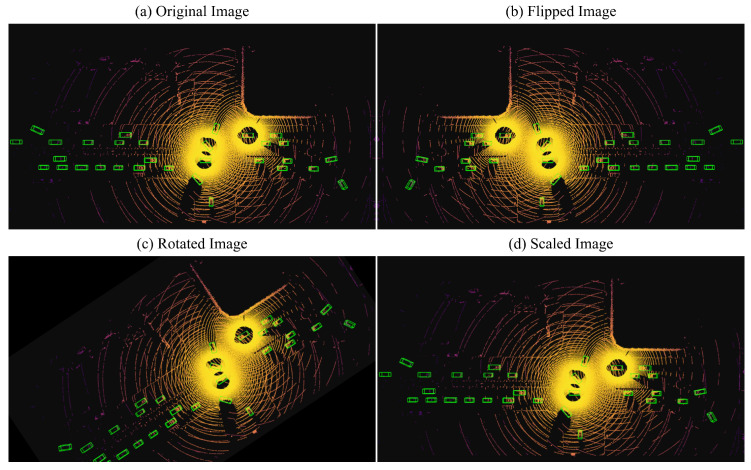
Data enhancement visualization results, where the green box is the ground truth. (**a**) shows the original data visualization, (**b**) shows the flipped visualization image, (**c**) shows the rotated visualization image, and (**d**) shows the scaled visualization image.

**Figure 9 sensors-24-02170-f009:**
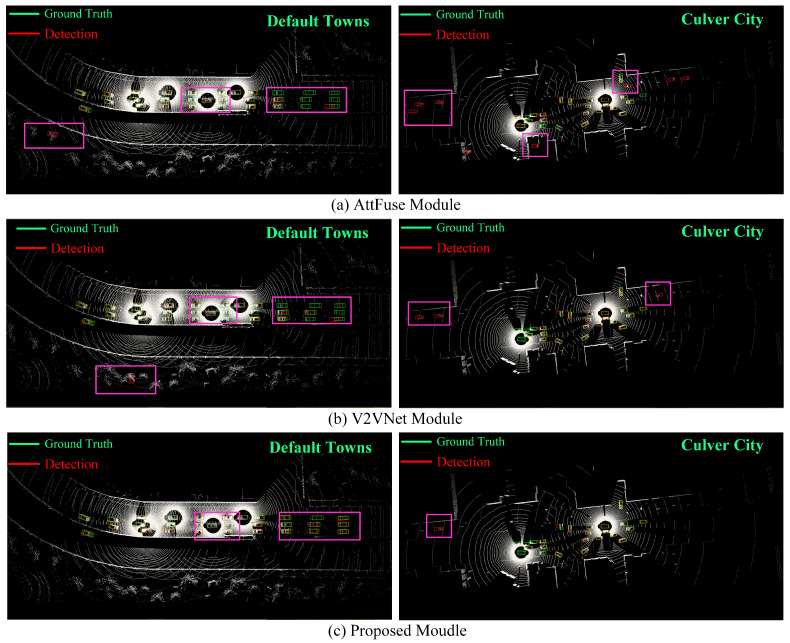
Comparison chart of detection results between mainstream models based on intermediate fusion and the proposed intermediate fusion model. (**a**) shows the detection results of the AttFuse model for Default Towns and Culver City. (**b**) shows the detection results of the V2VNet model. (**c**) shows the detection results of the proposed detection method.

**Figure 10 sensors-24-02170-f010:**
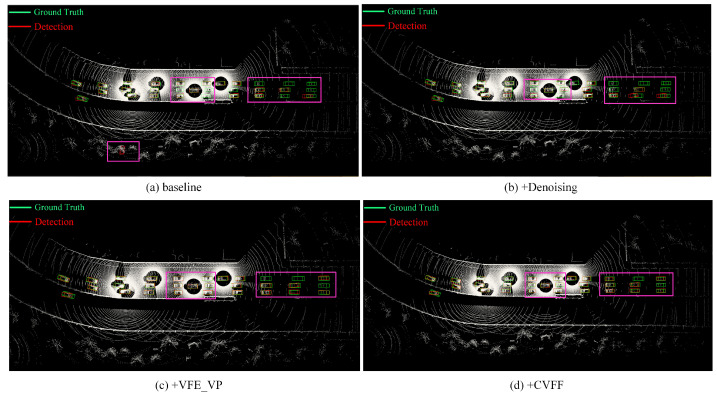
Schematic diagram of ablation experiment. (**a**) shows the detection results of the benchmark method of this paper in Default Towns. (**b**) shows the detection results after adding the point cloud denoising method, as described in Section 3.1. (**c**) shows the detection results after adding the feature extraction method of voxel point column fusion, as described in Section 3.2. (**d**) shows the detection results after adding the crossvehicle feature fusion module, as described in Section 3.4.

**Table 1 sensors-24-02170-t001:** Comparison of the AP values and model size for different methods.

Method	Default Towns		Culver City		Model Size (Mb)
	**AP@0.5**	**AP@0.7**	**AP@0.5**	**AP@0.7**	
No Fusion	49.1	38.3	40.6	26.7	18.2
Early Fusion	52.3	40.6	42.5	35.3	20.0
Late Fusion	59.6	42.5	49.4	39.7	19.5
F-Cooper [32]	61.7	49.8	53.7	44.5	35.3
Who2com [4]	62.0	50.5	54.1	44.2	37.4
AttFuse [33]	62.8	50.8	54.0	46.3	34.3
V2VNet [34]	63.3	51.6	54.5	45.8	36.8
HP3D-V2V (Ours)	67.4	56.5	58.8	50.5	35.0

**Table 2 sensors-24-02170-t002:** Evaluation results of ablation experiments.

Method	Default Towns			Culver City	
	**AP@0.5**	**AP@0.7**		**AP@0.5**	**AP@0.7**
Baseline	SECOND	60.4	48.7	55.3	45.1
	PointPillar	61.5	49.2	54.5	44.4
+Denoising	SECOND	61.7	49.6	56.0	45.8
	PointPillar	63.1	54.5	55.3	46.4
+VFE_VP	SECOND	–	–	–	–
	PointPillar	65.5	55.0	56.7	47.5
+CVFF	SECOND	64.3	53.1	56.1	47.2
	PointPillar	67.4	56.5	58.8	50.5

## Data Availability

No new data were created or analyzed in this study. Data sharing is not applicable to this article.

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
