# Peer review of "HP3D-V2V: High-Precision 3D Object Detection Vehicle-to-Vehicle Cooperative Perception Algorithm"

_sensors, 2024, doi:10.3390/s24072170_

Round 1

Reviewer 1 Report

Comments and Suggestions for Authors

This paper proposes a robust intermediate fusion approach to enhance feature selection and information aggregation, and improve perception accuracy in connected autonomous vehicles (CAVs). The topic selection of the paper is innovative and pioneering. The paper presents a clear overall idea and the sentences are generally smooth. The research work is challenging and the workload is substantial. However, there are some shortcomings that need to be addressed.

1. n Section 3.2, the proposed method for splicing in the Feature Learning Network is too rigid. It is recommended to provide a detailed explanation of the differences and advantages between voxelization and pillarization of BEV, and to use visual examples to illustrate the concept.

2. In Section 3.3, the proposed feature pyramid network based on standard and transposed convolutions aims to achieve near density and far sparsity properties. The direct splicing method is chosen to aggregate feature maps from different layers. The advantages of this method are not explained.

3. In the second half of Figure 6, the IOU loss function is chosen instead of more advanced options such as GIOU or Wise IOU. The benefits of this choice are not explained.

4. In Chapter IV's experimental section, Table 1 compares too few models. To make the experiment more comprehensive, it is suggested to compare more recent models.

5. In section 4.2.5, enhanced visualization results can be listed to improve the article's completeness.

6. The innovation part of the abstract needs further refinement to highlight the study's innovation, and the key words need to be revised.

Comments on the Quality of English Language

Minor editing of English language required

Author Response

We would like to thank the reviewer for spending his/her time to assess the paper, and make some very constructive and detailed informative comments provided in the review, and we have meticulously addressed each point in the attached document.

Reviewer 2 Report

Comments and Suggestions for Authors In this paper, a High-Precision 3D Object Detection Vehicle-to-Vehicle  Cooperative Perception Algorithm is proposed. This proposed method achieves an optimal balance between accuracy and bandwidth requirements.    Some minor problems should be revised, like    The title 3 . Proposed HP3D-V2V Algorithm    should be ' HP3D-V2V Algorithm' Comments on the Quality of English Language 3.4.2   The first sentence 'we introduce an encoder-decoder architecture...'  'we'  should be upper letter  We.  

Author Response

We would like to thank the reviewer for his/her careful examining work and the time spent. The constructive and detailed feedback provided in the review is immensely appreciated, and we have meticulously addressed each point in the attached document.

Reviewer 3 Report

Comments and Suggestions for Authors

1 The study propose a highly accurate  cooperative perception algorithm designed for 3D target detection between vehicles. The proposed tests shown the best achievement for the tested dataset (benchmark). The achievements in CNN architectures.

2 The introduction should contain aim of the paper, expected that aim formulation will be corresponds to the paper title.

3 The 4-th contribution point "To train, validate, and evaluate the performance of the proposed algorithm, The OPV2V dataset is used to validate the effectiveness and robustness of the algorithm in various scenarios and conditions" does not have any real novelty, For instance authors can find another papers with results tests on  OPV2V here https://paperswithcode.com/dataset/opv2v

4 The related work section does not contain any results after 2022, and should be extended with them.

5 The results of the paper tab2 are less than sota in https://paperswithcode.com/sota/3d-object-detection-on-opv2v in the conditions seems to be the same, authors should explain this point.

Also SOTA comparison does not contains results of 2023-2024 years for the tested dataset. This point should be also clarified.

6 The conclusion describe main items of the proposed architecture of the 3D target detection neural network, should include all contribution points.

7. Please describe how the conclusions are or are not consistent with the evidence and arguments presented.  

8. The mentioned terms need to be clarified :in the line 477 term "bandwidth" , in the line 479 "robust performance" - my opinion these terms  was not evaluated. 

9. Please also indicate if all main questions posed were addressed and by which specific experiments. - authors should discuss more in depth contribution points.

10. In general the paper is well written and after fix of notations seems to be on the level of research area.

Author Response

We extend our sincere gratitude to the reviewer for dedicating valuable time to evaluate our paper. The constructive and detailed feedback provided in the review is immensely appreciated, and we have meticulously addressed each point in the attached document.

Round 2

Reviewer 1 Report

Comments and Suggestions for Authors

Thank you for the author's careful revision.

Comments on the Quality of English Language

Minor editing of English language required